# The Effects of Rhythm Jump Training on the Rhythmic Reproduction Ability in Jumping and Agility in Elementary School Soccer Players

**DOI:** 10.3390/children12020133

**Published:** 2025-01-26

**Authors:** Yudai Kato, Hiroyuki Watanabe, Masashi Kawabata, Noritaka Mamorita, Ryota Muroi, Yukiyasu Tsuda, Yuto Uchida, Yusuke Tsuihiji, Koharu Mogi, Yuto Watanabe, Yuto Sano, Naonobu Takahira

**Affiliations:** 1Department of Therapy for Sports and Musculoskeletal System, Kitasato University Graduate School of Medical Sciences, Sagamihara 252-0373, Japan; zama_reha_pt@jin-ai.or.jp (Y.K.); mkawaba@kitasato-u.ac.jp (M.K.); mamorita@kitasato-u.ac.jp (N.M.); uchida.yuto@st.kitasato-u.ac.jp (Y.U.); watanabe.yuto@st.kitasato-u.ac.jp (Y.W.); sano.yuto@st.kitasato-u.ac.jp (Y.S.); takahira@med.kitasato-u.ac.jp (N.T.); 2Department of Rehabilitation, Zama General Hospital, Zama 252-0011, Japan; 3Department of Rehabilitation, Kitasato University School of Allied Health Sciences, Sagamihara 252-0373, Japan; 4Department of Medical Informatics, Kitasato University School of Allied Health Sciences, Sagamihara 252-0373, Japan; 5Department of Sports Medicine, St. Marianna University School of Medicine, Kawasaki 216-8512, Japan; ryota.muroi@marianna-u.ac.jp; 6Sports Training Association of Rhythm, Okayama 708-0062, Japan; star1130@ymail.ne.jp; 7Department of Rehabilitation, Sagamihara Kyodo Hospital, Sagamihara 252-5188, Japan; 8Department of Rehabilitation, Keiyu Orthopedic Hospital, Tatebayashi 374-0013, Japan; tsuihiji.yusuke@st.kitasato-u.ac.jp; 9Department of Rehabilitation, Nippon Koukan Hospital, Kawasaki 210-0852, Japan; mogi.koharu@st.kitasato-u.ac.jp

**Keywords:** rhythmic ability, rhythm jump training, agility

## Abstract

** Background/Objectives**: This study aimed to elucidate the effect of rhythm jump training on the rhythm and motor abilities of elementary school students to provide insights into its potential benefits for their physical performance and coordination. **Methods**: A non-randomized controlled trial was conducted involving 101 elementary school students (grades 1–6) attending a soccer school. Participants were divided into a rhythm jump group (n = 51, age: 7.5 years, height: 126.0 cm, weight: 25.7 kg) and a control group (n = 39, age: 8.0 years, height: 128.8 cm, weight: 26.5 kg) based on their practice venue. The rhythm jump group engaged in 10 min of rhythm jump at the beginning of soccer practice once a week over 8 weeks (intervention period), while the control group continued regular soccer training. Measurements included rhythmic reproduction ability during jumps, Pro Agility Test (PAT) values, and Reactive Strength Index (RSI) scores, assessed before and after the intervention period. Rhythmic reproduction ability was measured by comparing the data of the timing of jumps to 4- and 8-beat audio tracks. These were analyzed using repeated-measures analysis of variance, with significance set at *p* < 0.05. **Results**: Significant interactions were observed between 8-beat rhythmic reproduction ability and PAT values. In 8-beat rhythm deviation, a significant decrease was observed in the rhythm jump group (0.048 s) compared to that in the control group (0.013 s) (*p* < 0.01). PAT time significantly decreased in the rhythm jump group (0.18 s) compared to the control group (−0.25 s) (*p* < 0.01). There was no observed interaction between 4-beat rhythmic reproduction ability and RSI. **Conclusions**: This study revealed that rhythm jump training can be effective even with short sessions and infrequent practice, emphasizing its efficiency. The short-term rhythm jump intervention improved the 8-beat rhythmic reproduction ability and agility of elementary school students.

## 1. Introduction

Since 2015, rhythm jump training (also known as rhythm jump) has emerged as an enjoyable training method for children, contributing to the improvement of motor skills. It involves jumping to musical rhythms, which serves as an exercise to enhance motor skills and boost motivation for physical activity.

One of the characteristics of rhythm jump is the repetition of jumps. Jumping is regarded as important because it plays a significant role in children’s health as well as in the development of leg strength and balance [1,2]. Additionally, exercises involving continuous jumping, such as rope skipping, have been reported to improve physical abilities such as agility [2], making them a recommended activity for children.

Additionally, rhythm jump is characterized by jumping to musical rhythms. It has been reported that rhythmic activities, such as finger tapping, are associated with brain function [3]. Even without physical movement, perceived rhythmic actions activate the brain regions associated with movement, including the cerebellum, the primary motor cortex, and the premotor cortex [4]. Training in beat perception, which involves recognizing regular time intervals when listening to music [5], has been reported to make movements more rhythmic [5,6]. This indicates a close relationship between rhythm and movement. Rhythmic ability refers to the capacity to synchronize or reproduce rhythm [7]. Rhythmic abilities are believed to enhance motor performance and can be improved through training [8,9]. The “Interactive Metronome”, a training method that requires performing different actions synchronized with metronome sounds, has been reported not only to improve motor performance but also to enhance rhythmic ability [10,11].

The literature indicates that rhythm jump could influence both motor skills and rhythmic ability. However, the impact of rhythmic jumps on rhythmic abilities is still uncertain. Therefore, this study aimed to clarify the effects of rhythm jump on the rhythmic ability and motor skills of elementary school soccer players. We hypothesized that performing rhythm jump once a week over an 8-week period would improve rhythmic ability, jumping ability, and agility.

## 2. Materials and Methods

### 2.1. Study Design and Participants

This study was a non-randomized controlled trial conducted between April 2023 and August 2023. The study involved 101 elementary school students (grades 1–6) attending soccer schools. Allocation was conducted across five practice venues in soccer schools. The participants were recreational-level players in training stages. Two venues were assigned to the rhythm jump group (51 participants), while the control group (46 participants) was assigned to three venues (Figure 1).

The inclusion criteria were healthy elementary school students with exercise habits. Individuals experiencing pain during the movement of the lower limbs or with missing data on the measurement items were excluded.

Participants and their proxies were provided with written and verbal explanations regarding the purpose and methods of the study, voluntary participation, assurance of no disadvantages for refusal, and protection of personal information. The proxies provided written informed consent after receiving explanations of this research. The study was approved by the Ethics Committee of our institution, which ensured the anonymity of the participants (approval number: 2022-029).

### 2.2. Allocation and Blind

We employed an open-label assessor-blind approach, wherein assessors were unaware of the assignment of participants to either the rhythm jump or control groups. The personnel involved in assigning participants were the same as those conducting the interventions. Assignments were determined according to the practice days and the varying sizes of school clusters at each practice venue.

### 2.3. Intervention

The rhythm jump group incorporated rhythm jump into the first 10 min of their 60-min school practice sessions. The intervention lasted for 8 weeks, with sessions conducted once a week for a total of eight sessions. During these sessions, participants engaged in six types of rhythm jump exercises (Figure 2).

### 2.4. Procedure

Pre-intervention measurements included collecting basic information (age, height, weight) on paper and assessing rhythmic reproduction ability (4- and 8-beat rhythmic reproduction ability), agility (Pro Agility Test: PAT), and jump ability (Reactive Strength Index: RSI). All measurements were conducted at the soccer school venues after practice and administered by assessors (excluding interveners). After pre-intervention measurements, participants were allocated to either the rhythm jump group or the control group. The rhythm jump intervention commenced one week after the pre-intervention measurements for the rhythm jump group. The intervention lasted 8 weeks, with interveners conducting sessions once a week for 10 min, totaling 8 sessions. Post-intervention measurements were conducted by assessors one week after the final intervention session.

### 2.5. Outcome Measurements

#### 2.5.1. Rhythmic Reproduction Ability

Rhythmic reproduction ability was defined as the ability to reproduce rhythms heard while jumping. Through headphones, participants listened to audio recordings of 4-beat plus 8-beat rhythms generated using GarageBand^®^ (Apple Inc., Cupertino, CA, USA). The 4-beat rhythm contained four quarter notes per measure, while the 8-beat rhythm consisted of eight eighth notes per measure. The audio recordings were set to a tempo of 120 beats per minute (BPM), with a time interval of 0.5 s for the 4-beat rhythm and 0.25 s for the 8-beat rhythm. To ensure that the participants understood the task, they were asked to reproduce the rhythms they heard by tapping their fingers (finger taps). If the evaluator determined that the rhythm could not be replicated through finger tapping, the participants were asked to listen to the audio recordings again. Once the participants successfully replicated the rhythm through finger tapping, they stood on a Mat Switch® (Sports Sensing Co., Ltd., Fukuoka, Japan) and reproduced the heard rhythm through jumping.

#### 2.5.2. Pro Agility Test (PAT)

PAT assesses agility by having the participants sprint 20 m while changing direction twice within a 10 m straight distance [12]. The course of the PAT is illustrated in Figure 3. The participants started with both feet astride the central line and their right hand touching the line [13,14]. Upon hearing the signal “go” from the examiner, participants sprinted to the right line, touched it, pivoted, sprinted to the left line, touched it, pivoted again, and sprinted through the central line, which marked the finish. The examiner stood at the central line and used a stopwatch to measure time. Stopwatch measurement methods have shown high inter-rater reliability, with an intraclass correlation coefficient of 0.99 [14]. PAT measurements were conducted on artificial turf commonly used in soccer practice venues. Participants completed the tests twice, and the fastest time recorded was used for analysis. If any situations arose during the measurement, such as falling or obvious deceleration requiring measurement discontinuation, a third measurement was performed.

#### 2.5.3. Reactive Strength Index (RSI)

RSI is used to quantify the performance of the stretch-shortening cycle (SSC) [15], where the transition from eccentric to concentric contraction occurs [16].

RSI is calculated using jump height (m) per ground contact time (s), as indicated in Equation (1) [17].(1)RSI=Jump Height/Ground Contact Time

Measurement was conducted using PUSH2.0^®^ (S & C Corporation, Kyoto, Japan). Participants wore the PUSH2.0^®^ by securing it around the waist with a belt and then wrapping an elastic bandage over it to fix it in place. After the task was demonstrated, verbal instructions were given to “jump as high and as fast as possible for 10 consecutive jumps”, and participants performed the task accordingly. The measurements were performed on a mat laid over the concrete, and the procedure was repeated twice. The maximum value obtained from the measurements was used for the analysis. Additionally, if the measurement had to be discontinued owing to factors such as falling or stopping jumps, a third measurement was conducted.

### 2.6. Rhythmic Reproduction Ability Analysis

The flow of measurement for rhythmic reproduction ability analysis is shown in Figure 4. The signal waveform of participants jumping on the Mat Switch^®^ was captured using the Measurement Control Application^®^. The waveform data were imported into a personal computer, where the *x*-axis represents time (s) × sampling frequency (500 Hz), and the *y*-axis represents the pressure on the mat (v) in a two-dimensional graph.

Using MATLABR2022b^®^ (The MathWorks Inc., Natick, MA, USA), the time series pressure signal was subjected to continuous wavelet transform (Morse wavelet, time-bandwidth product 60, symmetry parameter 3, frequency resolution per octave 40). This was represented in a scalogram where the *x*-axis represented time (s), the *y*-axis represented frequency (Hz) representing the speed of movement, and the *z*-axis represented the absolute value of the wavelet coefficients representing the amount of frequency (hereafter referred to as amplitude) (au).

In theory, since the movements during measurement consisted only of 4-beat and 8-beat rhythm components, it was assumed that there would be only two frequencies where the amplitude would reach its maximum in the scalogram. To exclude the influence of frequency components related to artifacts and other factors, only values exceeding 80% of the maximum amplitude were summed over the measurement time, and the relationship between frequency and amplitude sum was obtained. From these results, among these peaks, the one with the highest total amplitude sum (exceeding 80%) across all time points was designated as Peak 1, and the peak with the largest trough and rise from Peak 1 was designated as Peak 2. The frequency of the maximum point close to 2 Hz was considered as the participant’s 4-beat rhythm, and the frequency of the peak close to 4 Hz was considered as the participant’s 8-beat rhythm.

The time difference between the intervals of the audio source and the intervals of the jumps was measured as the rhythm deviation. The deviation of the 4-beat rhythm was considered as the 4-beat rhythmic reproduction ability, whereas the deviation of the 8-beat rhythm was considered as the 8-beat rhythmic reproduction ability. A smaller deviation indicates higher rhythmic reproduction ability. The deviations for the 4-beat and 8-beat rhythms were calculated by substituting the 4-beat and 8-beat rhythms (Hz) into time (s) and using the following formula for analysis.(2)$$The\;deviation\;of\;a\;4-beat\;rhythm=\left|\;4-beat\;rhythm(s)-0.5(s)\right|$$(3)$$The\;deviation\;of\;a\;8-beat\;rhythm=\left|\;8-beat\;rhythm(s)-0.25(s)\right|$$

### 2.7. Statistical Analysis

The sample size was calculated using G*Power software (v3.1.9.2, University of Kiel, Germany) with the following parameters: power (1 − β) = 0.9, effect size (Cohen’s f) = 0.25, and significance level (α) = 0.05. As a result, it was estimated that 46 participants would be required for this study. Analysis was conducted separately for each allocated group based on the “intention to treat” principle. The Mann–Whitney U test was performed to assess the differences in baseline information between the rhythm jump and control groups. A repeated-measures two-way analysis of variance was conducted for the 4-beat rhythmic reproduction ability, 8-beat rhythmic reproduction ability, RSI, and PAT, with intervention (rhythm jump group vs. control group) and time (pre-intervention vs. post-intervention) as factors. Mauchly’s test of sphericity was performed, and if the assumption of sphericity was violated, the *p*-values were adjusted using the Greenhouse–Geisser epsilon correction. The significance level was set at *p* < 0.05. IBM SPSS Statistics version 22^®︎^ (IBM Corporation, Armonk, NY, USA) was used for statistical analysis. Cohen’s f was used to determine the magnitude of the response of the outcome to rhythm jump (effect size (ES) calculation with <0.1: small effect, <0.25: moderate effect, ≧0.4: large effect).

## 3. Results

Six participants from the rhythm jump group were absent due to personal reasons during the post-intervention assessment, and one participant reported exercise-induced pain during the post-intervention assessment, resulting in a final assessment of 44 participants. In the control group, six participants were absent during the post-intervention assessment, and one participant reported exercise-induced pain during the post-intervention assessment, resulting in a final assessment of 32 participants. There were no significant differences in age, height, or weight between the rhythm jump and control groups (Table 1). Within the rhythm jump group, 37 participants (84%) attended all eight intervention sessions, while seven participants (16%) attended seven sessions. The overall attendance rate for all the intervention sessions was 98%. The intervention period lasted 11 weeks in the intervention group and 11–13 weeks in the control group, owing to rain and scheduling constraints on soccer school practice days.

The 4-beat rhythmic reproduction ability did not show a significant interaction between the intervention and time (*p* = 0.55, F = 0.36, Cohen’s f = 0.07) (Figure 5a). However, the 8-beat rhythmic reproduction ability showed a significant interaction between intervention and time, indicating that the rhythm jump group had a significantly decreased rhythm deviation compared with the control group (*p* < 0.01, F = 10.86, Cohen’s f = 0.38) (Figure 5b).

Similarly, PAT displayed a significant interaction between intervention and time, with the rhythm jump group showing a significant decrease in PAT time compared to the control group (*p* < 0.01, F = 18.53, Cohen’s f = 0.50) (Figure 6).

RSI did not show a significant interaction between intervention and time (*p* = 0.22, F = 1.52, Cohen’s f = 0.14) (Figure 7).

## 4. Discussion

This study is the first to examine the impact of short-term rhythmic jump interventions on the rhythmic and motor abilities of elementary school students. The results revealed that the short-term rhythm jump intervention improved the 8-beat rhythmic reproduction ability and agility of elementary school students. This study included 76 participants, which provided sufficient statistical power based on our power calculations. Therefore, the intervention effect of rhythm jump training on rhythmic reproduction ability and agility in this study is considered valid.

### 4.1. Rhythmic Reproduction Ability

Significant improvements were observed in the 8-beat rhythmic reproduction ability within the rhythm jump group compared with that in the control group. This improvement can be attributed to the complex nature of fast movements, which require more intricate adjustments of the agonist and antagonist muscles than slower movements, suggesting a greater difficulty in motor execution [18]. In rhythm jump, participants engage in complex tasks involving not only 4-beat rhythms but also a combination of 4- and 8-beat rhythms. Through repetitive interventions involving jumps incorporating 8-beat rhythms, it is plausible that the rhythm jump group improved their more challenging 8-beat rhythmic reproduction ability compared to the control group.

A contributing factor to the improvement in 8-beat rhythmic reproduction ability through rhythm jump is the activation of brain regions and the optimization of central nervous system circuits. Perceiving, memorizing, and reproducing rhythms involves activity in various brain regions, including the cerebellum, basal ganglia, primary motor cortex, premotor cortex, supplementary motor area, and pre-supplementary motor area [4,19,20,21]. Interactive Metronome has been reported to activate functional connections in the brain, including those involving the cerebellum [22], leading to improved rhythmic ability [10,11]. Notably, the cerebellum exhibited increased activity when perceiving irregular rhythms compared with the perception of regular rhythms [4,20]. The role of the cerebellum includes the control of voluntary movements, muscle tension [23], memory of time intervals [4], planning and preparation of movements [24], timing perception, and control of movement execution timing [25]. Given these insights, the repetitive performance of rhythm jumps involving 4-beat plus 8-beat rhythms may have improved 8-beat rhythmic reproduction ability through the activation of brain regions, including the cerebellum, and the optimization of central nervous system circuits.

Conversely, the intervention effect of rhythm jumps did not affect 4-beat rhythmic reproduction ability. McAuley et al. reported a deviation between target beats and tap movements when recalling a constant rhythm with an interval of 0.506 s between beats from 4-year-olds to adults [26]. They noted that after the ages of 6–7 years, there was no significant change in rhythm deviation, suggesting that the 0.5-s interval of the 4-beat rhythm used in this study might have been easily reproducible for the participants.

McAuley et al. reported rhythm deviations of 0.021 s, 0.027 s, and 0.002 s during tapping movements in children aged 6–7, 8–10, and 10–12 years, respectively [26]. In this study, the rhythm deviations for the 4-beat rhythm among elementary school students were 0.05 s in the rhythm jump group and 0.06 s in the control group, indicating greater rhythm deviations compared to the previous study. A possible reason for this discrepancy is that the previous study measured rhythm using a tapping motion, whereas our study employed jumping movements. Jumping movements require full-body coordination involving both the upper and lower limbs [27]. Many sports, such as soccer, demand coordination between the upper and lower limbs. Therefore, measuring rhythm ability using jumping movements in this study is considered to better reflect the rhythm abilities necessary for sports. However, the choice of motion for rhythm ability measurement requires further discussion in the future.

### 4.2. PAT

PAT showed a significant decrease in time within the rhythm jump group compared to the control group. Since the study did not measure the time for a 20 m sprint, it is unclear whether rhythm jump influenced overall running ability or directional change speed. According to Young et al., factors such as foot placement, stride adjustment during acceleration or deceleration, and posture significantly affect directional speed change [28]. The rhythm jump intervention may have improved the ability to perform precise jumps, particularly concerning the 8-beat rhythmic reproduction ability. Improvements in executing precise movements, such as detailed jumping, could affect actions such as acceleration and deceleration during directional changes, potentially enhancing directional change speed. Consequently, the improvement in the 8-beat rhythmic reproduction ability suggests a possible influence on overall movement performance.

Zachary et al. reported that the average time for male athletes aged 6–9 years was 6.43 s, while it was 5.96 s for those aged 10–11 years [29]. In this study, the average time for elementary school students in the rhythm jump group was 7.34 s, indicating slower times compared to previous research. This suggests that rhythm jump may be effective for elementary school students with lower baseline abilities, but its effectiveness for those with higher baseline abilities requires further investigation.

### 4.3. RSI

RSI did not show any intervention effects from the rhythm jump. RSI is influenced by the development of muscles and neuromuscular systems [30], and reportedly undergoes significant development during adolescence [31]. The performance of the SSC has been suggested to be influenced by age, growth, and development, independent of training interventions [32,33]. Lloyd et al. conducted plyometric training for 9-year-old boys twice a week for 4 weeks and found no significant change in RSI [33]. This suggests that in elementary school children, RSI is greatly influenced by the growth-related development of muscles and neuromuscular systems, and the effects of rhythm jumps may not have been reflected in this measure.

### 4.4. Limitations

This study has three limitations. Firstly, participants were limited to elementary school students with exercise habits, potentially affecting the ability to generalize the findings. Future research should investigate the effect of rhythm jump training on elementary school students without exercise habits and other age groups. Secondly, variability in trainability and exercise difficulty based on grade levels posed a limitation. Although rhythm jump exercises can be tailored to different age groups and skill levels, this study standardized the intervention across all grade levels. Future studies should investigate the effects of changing the intervention exercises to accommodate different grade levels. Thirdly, the lack of verification of long-term effects is another limitation. This study demonstrated the short-term effectiveness of the intervention (10 min once a week for 11 weeks), but its long-term impact remains unexplored. It is essential to assess the long-term intervention effects and observe how interventions during childhood may influence future motor performance.

## 5. Conclusions

After eight sessions of rhythm jump training spanning 11 weeks, elementary school students showed improvement in both 8-beat rhythmic reproduction ability and agility. Furthermore, this study demonstrated that the rhythm jump is an effective intervention, even with short durations and low frequencies, emphasizing its efficiency.

## Figures and Tables

**Figure 1 children-12-00133-f001:**
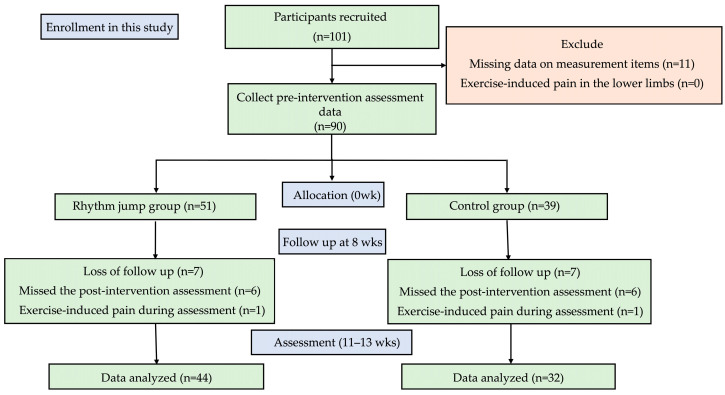
An outline of the study.

**Figure 2 children-12-00133-f002:**
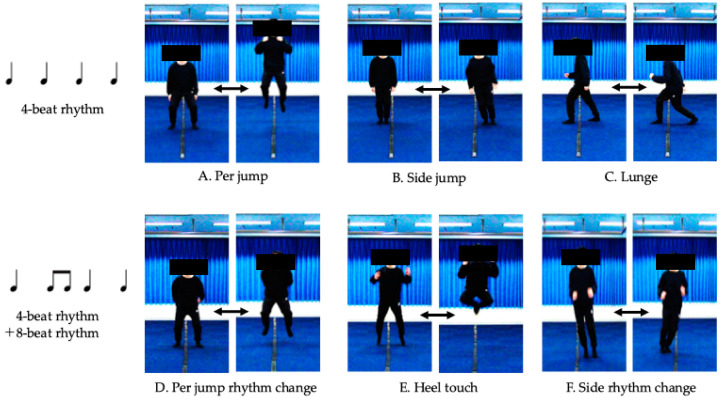
The intervention regimen. The first three exercises (**A**–**C**) were performed to a 4-beat rhythm, while the three latter exercises (**D**–**F**) combined a 4-beat rhythm with an 8-beat rhythm. The rhythm jump intervention was conducted by an individual with the Diffuser qualification certified by the Sports Rhythm Training Association. Meanwhile, the control group participated in regular soccer school practices throughout the 8-week intervention period.

**Figure 3 children-12-00133-f003:**
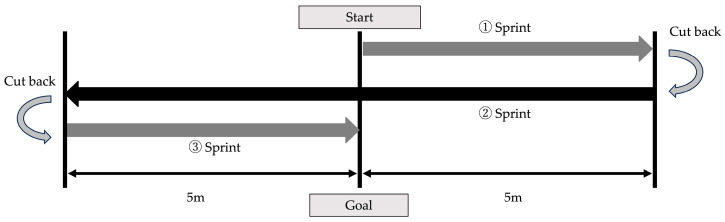
Measurement of Pro Agility Test.

**Figure 4 children-12-00133-f004:**
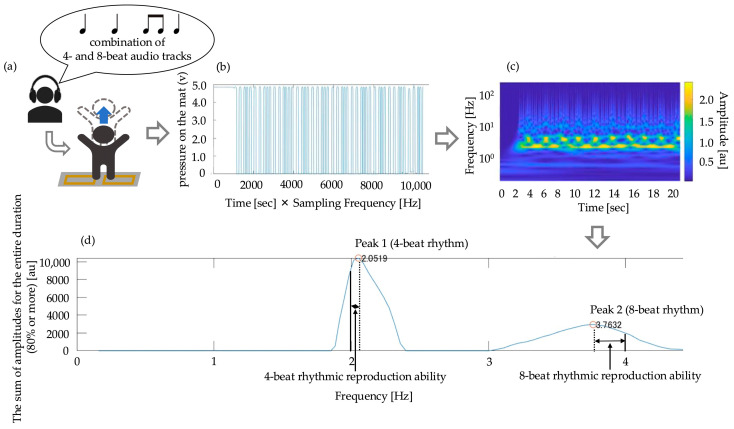
Flow from rhythmic reproduction ability measurement to analysis. (**a**) Listening to a combination of 4- and 8-beat audio tracks and then replicating the rhythm through jumping on a mat switch; (**b**) the waveform data from the mat switch where the *x*-axis represents time (s) × sampling frequency (500 Hz), and the *y*-axis represents the pressure on the mat switch (v) in a two-dimensional graph; (**c**) continuous wavelet transform for extraction of 4- and 8-beat rhythm; (**d**) calculation of the rhythm deviation (rhythmic reproduction ability). Solid line: reference frequency (2 and 4 Hz) of the audio source. Dotted line: the peak of frequency of the participant’s jump performance. The frequency of the maximum point close to 2 Hz was considered as the participant’s 4-beat rhythm, and the frequency of the peak close to 4 Hz was considered as the participant’s 8-beat rhythm. Between the solid and dotted lines: the time difference between the intervals of the audio source and the intervals of the jumps was measured as the rhythm deviation (rhythmic reproduction ability). 4-beat rhythmic reproduction ability was set to the difference between peak 1 and 2 Hz, whereas 8-beat rhythmic reproduction ability was set to the difference between peak 2 and 4 Hz.

**Figure 5 children-12-00133-f005:**
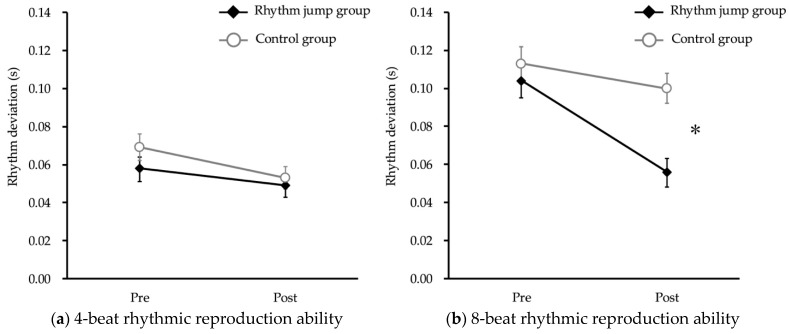
Rhythmic reproduction ability. The asterisk indicates a significant interaction between intervention (rhythm jump group vs. control group) and time (pre-intervention vs. post-intervention).

**Figure 6 children-12-00133-f006:**
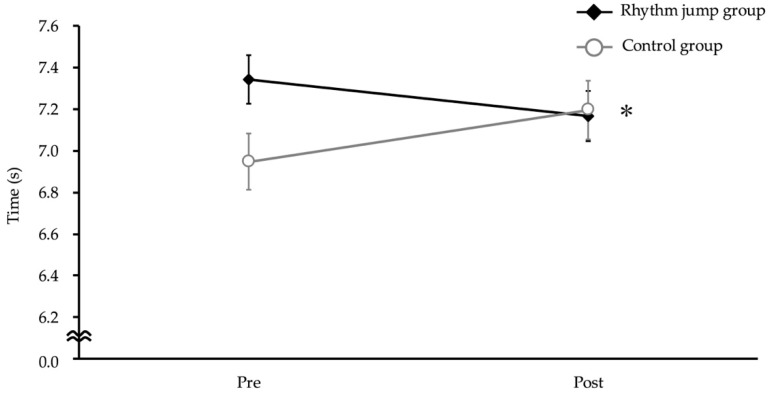
Pro Agility Test. The asterisk indicates a significant interaction between intervention (rhythm jump group vs. control group) and time (pre-intervention vs. post-intervention).

**Figure 7 children-12-00133-f007:**
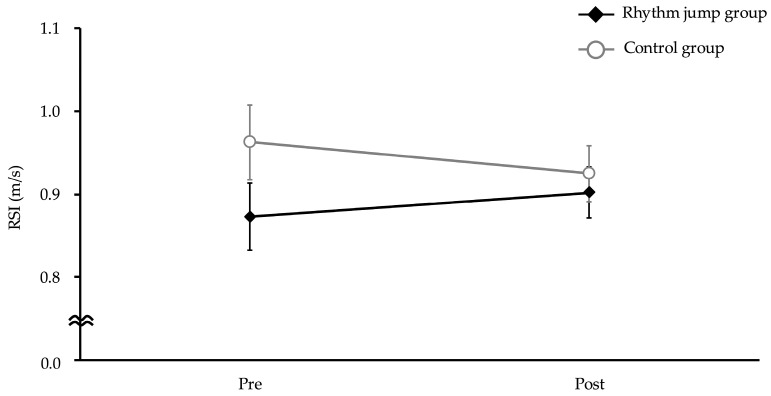
Reactive Strength Index.

**Table 1 children-12-00133-t001:** Details of the study participants.

	Rhythm Jump Group	Control Group	*p* Value
Age (years)	7.5 [6.0–9.0]	8.0 [7.0–8.0]	0.11
Height (cm)	126.0 [118.9–133.0]	128.8 [124.5–134.3]	0.17
Weight (kg)	25.7 [21.5–29.4]	26.5 [23.4–28.7]	0.42

Data displayed as median [interquartile range].

## Data Availability

The data presented in this study are available on request from the corresponding author.

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
