# Peer review of "The Effects of Rhythm Jump Training on the Rhythmic Reproduction Ability in Jumping and Agility in Elementary School Soccer Players"

_children, 2025, doi:10.3390/children12020133_

Round 1
Reviewer 1 Report
Comments and Suggestions for Authors
Dear authors,
The aim of the study is very important because improving children's motor skills will allow them to have more chances of continuing to do physical exercise in the future and have a healthier adult life. In addition, jumping is always a motor skill where children often have many problems because it has not been developed correctly at the right time, and then they are afraid of jumping.
However, there is a lot of information that is not provided in the introduction. For example, information could be provided on the importance of jumping, the drawbacks that jumping has in adolescents. That is, providing information to value your work. In addition, as this is a recent topic, it would be interesting to know what variables have been measured previously, to later assess those that you propose.
Regarding the results, it would be necessary to add asterisks to the figures so that it can be understood by itself. And thus, without needing the text, to know what results have shown significant differences.
Finally, in the discussion, although they provide a little more information, there is repetitive information regarding the introduction. Therefore, it would be necessary to add other studies that compare their results.
Best regards.
Author Response
However, there is a lot of information that is not provided in the introduction. For example, information could be provided on the importance of jumping, the drawbacks that jumping has in adolescents. That is, providing information to value your work. In addition, as this is a recent topic, it would be interesting to know what variables have been measured previously, to later assess those that you propose.
Response: Thank you for your feedback. I have added information on the importance of jumping in adolescents and the gaps in existing knowledge.
Additional note: “One of the characteristics of rhythm jump is the repetition of jumps. Jumping is regarded as important because it plays a significant role in children's health, as well as in the development of leg strength and balance [1-2]. Additionally, exercises involving continuous jumping, such as rope skipping, have been reported to improve physical abilities like agility [3], making them a recommended activity for children.“ (Page 2, Lines 52–56)
Regarding the results, it would be necessary to add asterisks to the figures so that it can be understood by itself. And thus, without needing the text, to know what results have shown significant differences.
Response: Thank you for your comment. I have added asterisks to the figure. (Page 8, Lines 272–274, and 280–282)
Finally, in the discussion, although they provide a little more information, there is repetitive information regarding the introduction. Therefore, it would be necessary to add other studies that compare their results.
Response: Thank you for your feedback. The following sentence has been deleted.
Delete: “PAT involves a sprint covering a total of 20 meters, with two directional changes within a straight-line stretch distance of 10 meters.“ (Page 10, Lines 345–346)
Delete: “Activities such as the“(Page 9, Line 314)
Delete: “which entails synchronized movements to metronome sounds,“ (Page 9, Lines 312–313)
A new study was added to compare the results.
Additional note: “McAuley et al. reported rhythm deviations of 0.021 seconds, 0.027 seconds, and 0.002 seconds during tapping movements in children aged 6–7, 8–10, and 10–12 years, respectively [26]. In this study, the rhythm deviations for the 4-beat rhythm among elementary school students were 0.05 seconds in the rhythm jump group and 0.06 seconds in the control group, indicating greater rhythm deviations compared to the previous study. A possible reason for this discrepancy is that the previous study measured rhythm using a tapping motion, whereas our study employed jumping movements. Jumping movements require full-body coordination involving both the upper and lower limbs [27]. Many sports, such as soccer, demand coordination between the upper and lower limbs. Therefore, measuring rhythm ability using jumping movements in this study is considered to better reflect the rhythm abilities necessary for sports. However, the choice of motion for rhythm ability measurement remains that requires further discussion in the future. “(Page 10, Lines 329–341)
Additional note: “Zachary et al. reported that the average time for male athletes aged 6–9 years was 6.43 seconds, while it was 5.96 seconds for those aged 10–11 years [29]. In this study, the average time for elementary school students in the rhythm jump group was 7.34 seconds, indicating slower times compared to previous research. This suggests that rhythm jump may be effective for elementary school students with lower baseline abilities, but its effectiveness for those with higher baseline abilities requires further investigation.“ (Page 10, Lines 357–362)

Reviewer 2 Report
Comments and Suggestions for Authors
Effects of Rhythm Jump Training on The Rhythmic Reproduction Ability in Jumping and Agility in Elementary School Soccer Players REVIEW
This study aimed to elucidate the effect of rhythm jump training on the rhythm and motor abilities of elementary school students.
Clear experimental design represents the main strength of this study. On the other hand, lengths of treatment as well as representativeness of the sample are main weakness of this study.
The abstract is informative; I recommend authors to add some data about the sample in abstract.
Also, in line 25 we have some odd double space.
Keywords we’re not done carefully, this would have to be rewritten with a little more care and dedication so that it could serve the function it should have. As far as I'm concerned, this is not just a recommendation, but I consider it a mandatory part of preparing the next version of the work.
The introduction provides a clear theoretical base for asking research question. I would like to thank the authors for the precise and concise introduction, this is not often seen today. Very well done! However, perhaps they could consider adding hypotheses to the introduction, it is not mandatory, but it would help to follow and understand the study.
Method section gives enough information for repeating the study also data gathering and processing are appropriate to give clear answer on the research question. Having in mind mentioned shortcomings of the study, it is necessary to describe in more detail the method of designing the sample. A power analysis is also necessary. I would also recommend giving additional information like gender, sport level and experience. Table 1 should be mentioned in the participants section and moved up.
Data processing as well as results section lacks of effect size explication and evaluation. Also it is not clear why specific processing tests and procedures are used. It is not clear are prerequisites for its application such as normality, sphericity, homogenicy of the variance are meet. This is also mandatory.
To be able to evaluate are conclusions appropriate, information and discussion of statistical power and effect size is mandatory.
These findings expand our knowledge in the topic and can have clear practical application.
The references are appropriate.
I’ll be glad if my recommendations are helpful.
I have not additional comments.
Author Response
The abstract is informative; I recommend authors to add some data about the sample in abstract.
Also, in line 25 we have some odd double space.
Response: Thank you for your comment.
The summary includes the average values for age, height, and weight of the Rhythm Jump group and the Control group, as well as the number of participants in each group. (Page 1, Lines 28–29)
We apologize, but we could not find the double space in Line 25. Does it still remain? If it does, could you please point it out again?
Keywords we’re not done carefully, this would have to be rewritten with a little more care and dedication so that it could serve the function it should have. As far as I'm concerned, this is not just a recommendation, but I consider it a mandatory part of preparing the next version of the work.
Response: Thank you for your feedback. The items under "Keywords" have been updated.
rhythmic ability; rhythm jump training; agility (Page 1, Line 45)
The introduction provides a clear theoretical base for asking research question. I would like to thank the authors for the precise and concise introduction, this is not often seen today. Very well done! However, perhaps they could consider adding hypotheses to the introduction, it is not mandatory, but it would help to follow and understand the study.
Response: As you pointed out, adding the hypothesis has resulted in an improved background paragraph. Thank you very much.
Additional note: ”We hypothesized that performing rhythm jumps once a week over an 8 week period would improve rhythmic ability, jumping ability, and agility.” (Page 2, Lines 72–74)
Method section gives enough information for repeating the study also data gathering and processing are appropriate to give clear answer on the research question. Having in mind mentioned shortcomings of the study, it is necessary to describe in more detail the method of designing the sample. A power analysis is also necessary. I would also recommend giving additional information like gender, sport level and experience. Table 1 should be mentioned in the participants section and moved up.
Response: Thank you for your feedback.
Additional note: ”The sample size was calculated using G*Power software (v3.1.9.2, University of Kiel, Germany) with the following parameters: power (1−β) = 0.9, effect size (Cohen’s f) = 0.25, and significance level (α) = 0.05. As a result, it was estimated that 46 participants would be required for this study. ” (Page 7, Lines 233–237)
I added "sports level" as participant information.
Additional note: ”All participants in this study were at an amateur level.” (Page 2, Lines 80–81)
I moved Table 1 to the Participants section. (Page 2, Lines 97–98)
Data processing as well as results section lacks of effect size explication and evaluation. Also it is not clear why specific processing tests and procedures are used. It is not clear are prerequisites for its application such as normality, sphericity, homogenicy of the variance are meet. This is also mandatory.
Response: Thank you for your feedback. The explanation of effect size has been added to "Statistical Analysis" and the effect sizes for each measurement item have been included in the "Results". (Pages 7–8, Line 240, 246–248, 264, 267–268, 278, and 285)
Additional note ”Cohen's f were used to determine the magnitude of the response of the outcome to rhythm jump (effect size (ES) calculation with <0.1 = small effect, <0.25 = moderate effect, ≧0.4 =large effect).” (Page 7, Lines 246–248)
I added the conditions for applying the statistical analysis.
Additional note: ”A repeated-measures two-way analysis of variance was conducted…”, and “Mauchly's test of sphericity was performed. When the hypothesis was rejected, the significance probability was determined using the Greenhouse-Geisser epsilon correction.” (Page 7, Lines 240, and 243–245)
To be able to evaluate are conclusions appropriate, information and discussion of statistical power and effect size is mandatory.
Response: As you pointed out, adding statistical power helped emphasize the conclusion. Thank you very much.
Additional note: ”This study included 76 participants, which provided sufficient statistical power based on our power calculations. Therefore, the intervention effect of rhythm jump training on rhythmic reproduction ability and agility in this study is considered valid.” (Page 9, 292-295)

Round 2
Reviewer 1 Report
Comments and Suggestions for Authors
Dear authors,
Thank you for accepting the suggestions and incorporating them into the study. There is only one thing left to modify and that is in line 81, when it says that the participants have an amateur level, I consider that it cannot be described in that way due to their young age. And, the ideal is to say that they are players in training stages.
Best regards.
Author Response
Reviewer 1
Thank you for accepting the suggestions and incorporating them into the study. There is only one thing left to modify and that is in line 81, when it says that the participants have an amateur level, I consider that it cannot be described in that way due to their young age. And, the ideal is to say that they are players in training stages.
Thank you very much for your valuable feedback and for taking the time to review our revised manuscript. We greatly appreciate your constructive suggestions, which have helped us to improve the quality of our work.
Response: Regarding your comment on line 81, we completely understand your concern about the description of the participants as having an "amateur level" given their young age. We agree that the term could be misleading and have revised the text to reflect your suggestion. The revised sections are highlighted with a yellow marker.
"The participants are recreational level players in training stages." (Page 2, Lines 81)
Thank you again for your insightful feedback. Please let us know if there are any further adjustments required.
Reviewer 2 Report
Comments and Suggestions for Authors
IKt's ok now
Author Response
No comments were received from reviewer 2.